# Field-free spin-orbit switching of perpendicular magnetization enabled by dislocation-induced in-plane symmetry breaking

Yuhan Liang[1,10], Di Yi [1,10], Tianxiang Nan [2,10], Shengsheng Liu[1,3], Le Zhao [4,5], Yujun Zhang [6], Hetian Chen[1], Teng Xu[4,5], Minyi Dai[7], Jia-Mian Hu [7], Ben Xu [8], Ji Shi[9], Wanjun Jiang [4,5] ✉, Rong Yu [1,3] ✉ & Yuan-Hua Lin [1] ✉

Current induced spin-orbit torque (SOT) holds great promise for next generation magnetic-memory technology. Field-free SOT switching of perpendicular magnetization requires the breaking of in-plane symmetry, which can be artificially introduced by external magnetic field, exchange coupling or device asymmetry. Recently it has been shown that the exploitation of inherent crystal symmetry offers a simple and potentially efficient route towards field-free switching. However, applying this approach to the benchmark SOT materials such as ferromagnets and heavy metals is challenging. Here, we present a strategy to break the in-plane symmetry of Pt/Co heterostructures by designing the orientation of Burgers vectors of dislocations. We show that the lattice of Pt/Co is tilted by about 1.2° when the Burgers vector has an out-of-plane component. Consequently, a tilted magnetic easy axis is induced and can be tuned from nearly in-plane to out-of-plane, enabling the field-free SOT switching of perpendicular magnetization components at room temperature with a relatively low current density ($\sim 10^{11} A/m^2$) and excellent stability ($> 10^4$ cycles). This strategy is expected to be applicable to engineer a wide range of symmetry-related functionalities for future electronic and magnetic devices.

Modern magnetic-memory technology requires all-electric control of magnetization to realize devices with low energy consumption, high storage density and good thermal stability[1]. Recent advances have revealed that current-induced SOT provides an efficient means to electrically manipulate magnetic states[2–4]. In typical heavy metal (HM)/ferromagnetic metal (FM) heterostructures, a charge-current flowing in the plane gives rise to a spin-current along the out-of-plane direction according to spin-Hall scenario[5–7]. This spin-current is injected into FM layer, exerting a damping-like torque $\tau_{DL} \propto \mathbf{M} \times \boldsymbol{\sigma} \times \mathbf{M}$ and a field-like torque $\tau_{FL} \propto \boldsymbol{\sigma} \times \mathbf{M}$ on the magnetization $\mathbf{M}$, with $\boldsymbol{\sigma}$ being the spin polarization[7]. Considering the symmetry of heterostructure, the $\boldsymbol{\sigma}$ is in-plane and therefore a deterministic switching of perpendicular magnetization requires an additional in-plane symmetry breaking[8]. Previously, the in-plane symmetry breaking was artificially introduced by external magnetic fields[3–7], exchange coupling[9–13], and device asymmetry[14–19]. Recently, the exploitation of inherent crystal structure has been shown as a simple and efficient route to realize field-free SOT switching of perpendicular magnetization. However, so far,

[1]School of Materials Science and Engineering, Tsinghua University, Beijing, China. [2]School of Integrated Circuits and Beijing National Research Center for Information Science and Technology (BNRist), Tsinghua University, Beijing, China. [3]National Center for Electron Microscopy in Beijing, Tsinghua University, Beijing, China. [4]State Key Laboratory of Low-Dimensional Quantum Physics and Department of Physics, Tsinghua University, Beijing, China. [5]Frontier Science Center for Quantum Information, Tsinghua University, Beijing, China. [6]Institute of High Energy Physics, Chinese Academy of Sciences, Beijing, China. [7]Department of Materials Science and Engineering, University of Wisconsin–Madison, Madison, WI, USA. [8]Graduate School, China Academy of Engineering Physics, Beijing, China. [9]School of Materials and Chemical Technology, Tokyo Institute of Technology, Tokyo, Japan. [10]These authors contributed equally: Yuhan Liang, Di Yi, Tianxiang Nan. ✉e-mail: jiang_lab@tsinghua.edu.cn; ryu@tsinghua.edu.cn; linyh@tsinghua.edu.cn

demonstration has been limited to only a few unconventional materials with low crystal symmetry[20–23]. It is thus highly desirable to extend this symmetry-design approach to the benchmark SOT materials of HM/FM (such as Co, Pt, etc.), which remains a grand challenge since these materials usually show high crystal symmetry.

Here, we develop a strategy that uses dislocations to induce in-plane symmetry breaking in magnetic heterostructures. Dislocations, as one-dimensional defects prevailing in crystals, were commonly regarded as problematic for functional materials[24]. It has recently been shown that a proper engineering of dislocations could improve materials properties[25–27]. Dislocations disturb the perfect alignment of a crystal lattice, which potentially provides a microstructural tool to control the symmetry. Through rationally designing the orientation of Burgers vectors **B** of dislocations in model system of Pt/Co heterostructures on oxide templates, we realize the tilting of crystal lattice of Pt/Co by about 1.2° with respect to the underlying oxide templates. This lowering of crystal symmetry leads to a tilted magnetic easy axis that can be tuned from nearly in-plane to out-of-plane. Furthermore, a field-free SOT switching at room temperature is successfully demonstrated with a relatively low current density (~$10^{11}$ A/m²) and excellent stability (> $10^4$ cycles). Given that dislocations are prevailing in different classes of materials, this strategy should be generally applicable to manipulate the symmetry-related functionalities for high-performance devices.

## Design of dislocations for crystal tilting

In epitaxial heterostructures, the lattice mismatch between epitaxial layers and underlying substrates leads to the formation of dislocations near the interfaces in order to reduce the elastic energy. Dislocations introduce structural irregularities in perfect crystals, such as an extra half atomic plane introduced by an edge dislocation. This translational discontinuity, which is described by the Burgers vectors **B**, depends on

both the crystal structure of epitaxial layer and interface condition. In our model systems, Pt and Co in the bilayer heterostructure have face-centered-cubic (FCC) crystal structure[28,29], and therefore their Burgers vectors **B** are along $\frac{1}{2}$<110> directions[30]. We consider three main crystalline orientations of the heterostructures, e.g., (100)-, (110)- and (111)-orientations. As shown in Fig. 1a, for the (100)-orientation, there are two equivalent $\frac{1}{2}$<110> directions in the film plane. Therefore, the lattice mismatch between metal layers and underlying substrates can be fully accommodated by dislocations with in-plane Burgers vectors **B** (see Fig. 1b). As a result, the extra half atomic plane is normal to the underlying substrates and the crystal lattice is not expected to tilt as shown in Fig. 1c. Similar arguments are also valid for the (111)-orientation with three equivalent $\frac{1}{2}$<110> directions in the film plane (see Supplementary Fig. 1). By contrast, for the (110)-orientation (see Fig. 1d), there is only one $\frac{1}{2}$<110> direction within the film plane. Therefore, in order to fully accommodate the lattice mismatch across the interface, dislocations with Burgers vector that has an out-of-plane component are indispensable (see Fig. 1e). As schematically illustrated in Fig. 1f, this type of dislocation is expected to induce the tilting of the crystal lattice[31,32], which can be quantified by the angle $\theta_e$. As a result, the system becomes asymmetric relative to a 180° rotation along the normal direction of underlying substrates, leading to the breaking of in-plane symmetry.

To verify the above designing strategy, we fabricated textured Pt/Co bilayer heterostructures on NiO/MgO templates with the aforementioned three crystalline orientations (see Methods and Supplementary Fig. 2). Both MgO and NiO have the rock-salt structure with lattice constants (~4.2 Å) larger than that of textured Pt/Co films[28,33], which would lead to the formation of dislocations at the interfaces. The high-angle annular dark-field scanning transmission electron microscope (HAADF-STEM) is employed to characterize the atomic structure of the heterostructures. For the (100)-orientation heterostructure,

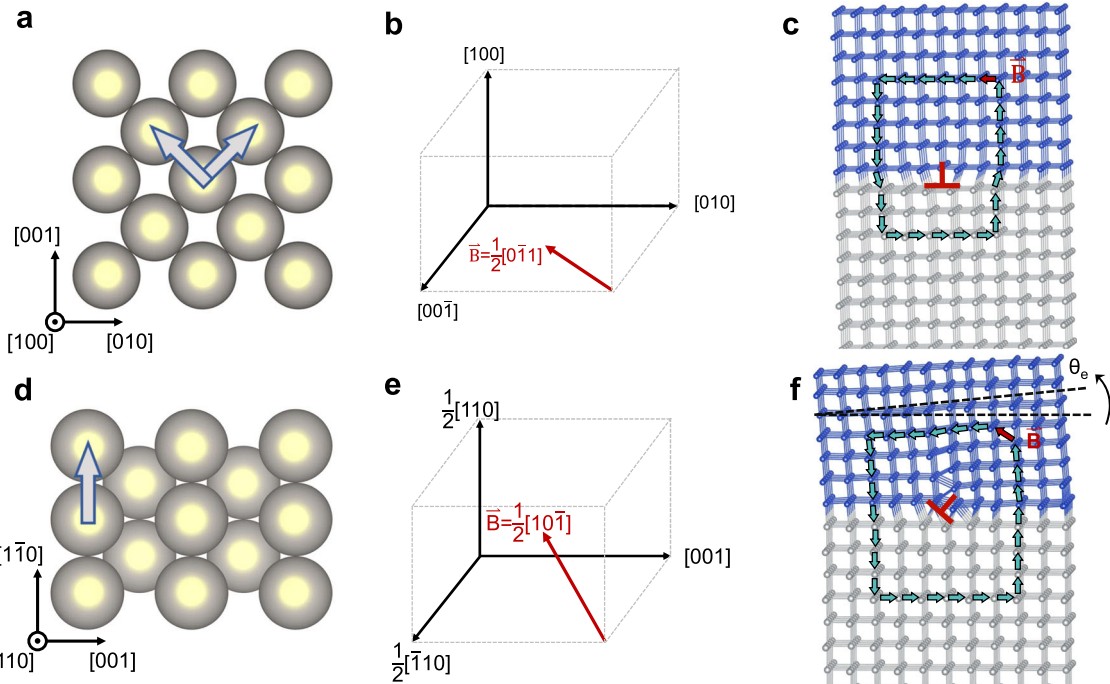

**Fig. 1 | The schematic of in-plane crystal symmetry breaking by dislocations. a** For the (100)-orientation, there are two $\frac{1}{2}$<110> lattice vectors in the film plane. The gold balls indicate the atoms constructing FCC lattice. Representative Burgers vector **B** is shown in **b**, resulting in **c**, dislocation with in-plane Burgers vector and no tilting of crystal lattice. **d** For (110)-orientation, there is only one $\frac{1}{2}$<110> lattice vector in the film plane. Therefore, Burgers vector **B** with out-of-plane component are required. Representative Burgers vector **B** is shown in **e**, results in **f** dislocation with an out-of-plane Burgers vector component and tilting of crystal lattice that is quantified by the angle $\theta_e$. The gray arrows indicate $\frac{1}{2}$<110> lattice vectors. Gray and blue balls indicate two distinct crystal lattices, and the Burgers circuits and Burgers vectors are indicated by green and red arrows, respectively. It is noted that FCC structure is not shown in Figs. **c** and **f** for better visualization.

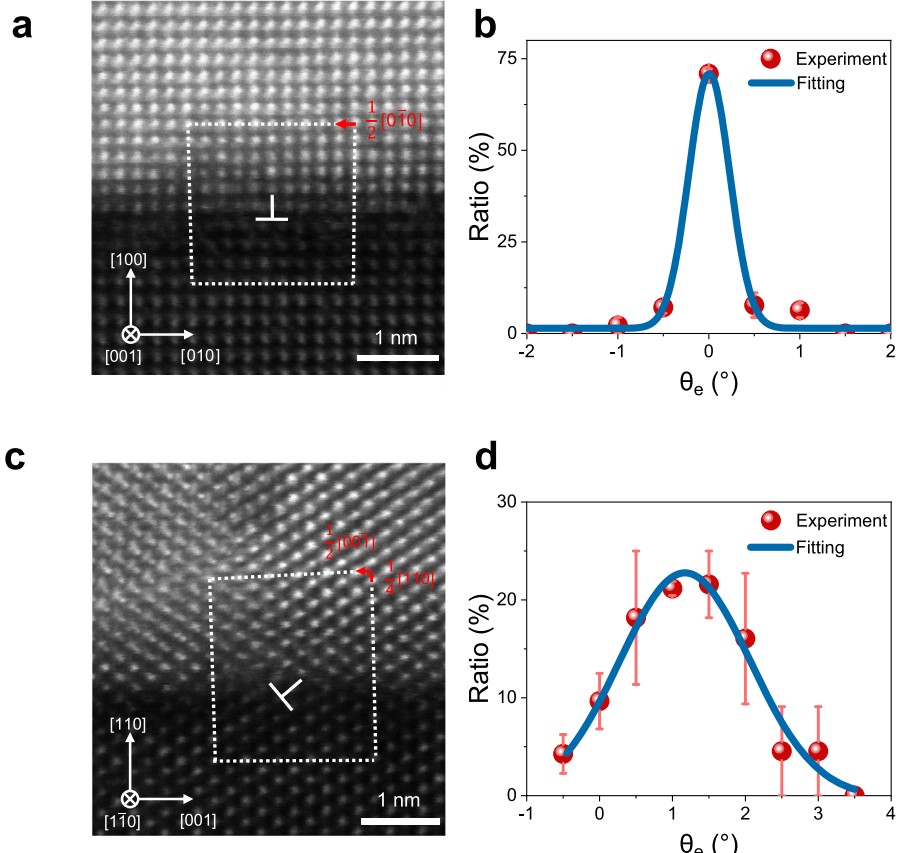

**Fig. 2 | The observation of in-plane crystal symmetry breaking by dislocations.**
**a** HAADF-STEM image of the (100)-orientation Pt(5)/Co(1.5)/NiO(20)/MgO hetero-
structure. Values in parentheses are thickness in nm. The statistical results of $\theta_e$ are
shown in **b**. **c** HAADF-STEM image of the (110)-orientation Pt(5)/Co(1.5)/NiO(20)/
MgO heterostructure. The statistical results of $\theta_e$ are shown in **d**. The white dash
boxes show the Burgers circuits, and the red arrows show the projected Burgers
vectors of dislocations. The symbol ⊥ indicates the dislocations. The blue line is the
gaussian fitting of the data. The error bars represent the standard deviation of $\theta_e$.

dislocations with only in-plane Burgers vector are observed (Fig. 2a),
revealed by drawing a Burgers circuit around the dislocation (white
dash line). The tilting angle $\theta_e$, as defined in Fig. 1f, is found to be close
to 0° by performing statistical analysis of the HAADF-STEM results
(Fig. 2b and Supplementary Fig. 3). By contrast, for the (110)-orienta-
tion heterostructure, dislocations with Burgers vectors that have both
in-plane and out-of-plane components are observed (Fig. 2c). The
crystal lattice of Co/Pt shows a clear tilting with respect to the oxide
templates, with $\theta_e$ around 1.2° via statistical analysis (Fig. 2d and
Supplementary Fig. 4) and arrays of dislocations at the interface (see
Supplementary Fig. 5). We note that the magnitude of $\theta_e$ and dis-
location density are in good agreement with our theoretical calcula-
tions based on the low-angle boundary model (see Supplementary
Note 1 and Supplementary Fig. 6). In addition, the crystal lattice tilting
is further detected by synchrotron-based X-ray diffraction, as shown in
Supplementary Fig. 7.

## Emergence of titled magnetic easy axis
Owing to the tilting of crystal lattice in the (110)-orientation hetero-
structures, the original mirror symmetry with respect to the (001)-
plane is broken, and thereby a tilted magnetic easy axis with an angle
$\theta_M$ is expected as shown in Fig. 3a. Figure 3b shows the magnetic
hysteresis results for the Pt/Co/NiO heterostructure with the (110)-
orientation, measured along different crystallographic directions. The
results exhibit a magnetic easy (hard) axis along the [001] ([$\bar{1}$10])
direction within the film plane. In addition, the finite remnant mag-
netization ($M_r$) along the out-of-plane ([110]) direction reveals the
tilting of the magnetic easy axis, showing the angle $\theta_M$ around 16° by

using $\theta_M = \arcsin(M_r/M_s)$ ($M_s$ is the saturation magnetization). We also
observe that the magnitude of $\theta_M$ decreases by increasing the Co layer
thickness ($t_{Co}$) in the heterostructures (see Supplementary Figs. 8 and
9), indicating the interfacial origin of the perpendicular magnetization
component.

To further verify the tilted anisotropy, we measure polar
magneto-optic Kerr effect (MOKE) and anomalous Hall effect (AHE)
signals, both of which mainly correspond to the perpendicular mag-
netization component. Through in-situ monitoring the change of polar
MOKE signals as sweeping magnetic field along the [001] direction, we
find that the perpendicular magnetization is reversibly switched by
reversing the in-plane magnetic field (Fig. 3c, Supplementary Figs. 10b
and 11), which strongly supports the presence of the tilted magnetic
easy axis in the (110)-orientation heterostructures (see Supplementary
Fig. 10a). This is further supported by the shift of polar MOKE hys-
teresis loop as sweeping out-of-plane field, when small in-plane bias
field is applied[34] (see Supplementary Figs. 12 and 13). Furthermore, we
also measure the polar angular ($\gamma$) dependence of Hall resistance ($R_{xy}$)
by rotating the external magnetic field (H = 100 mT) within the ($\bar{1}$10)
plane, as shown in Fig. 3d. The results reveal that the perpendicular
magnetization switches when the angle $\gamma$ is around 160° and 340°,
which are the characteristics of the tilted magnetic easy axis[35]. By
comparing experimental data in Fig. 3d to our simulation results, the
magnitude of $\theta_M$ is estimated to be around 20°, consistent with the
hysteresis results in Fig. 3b (see Supplementary Note 2 and Supple-
mentary Fig. 14). By contrast, this tilted magnetic easy axis is not
observed in heterostructures with (100)- and (111)-orientations, see
Supplementary Fig. 15.

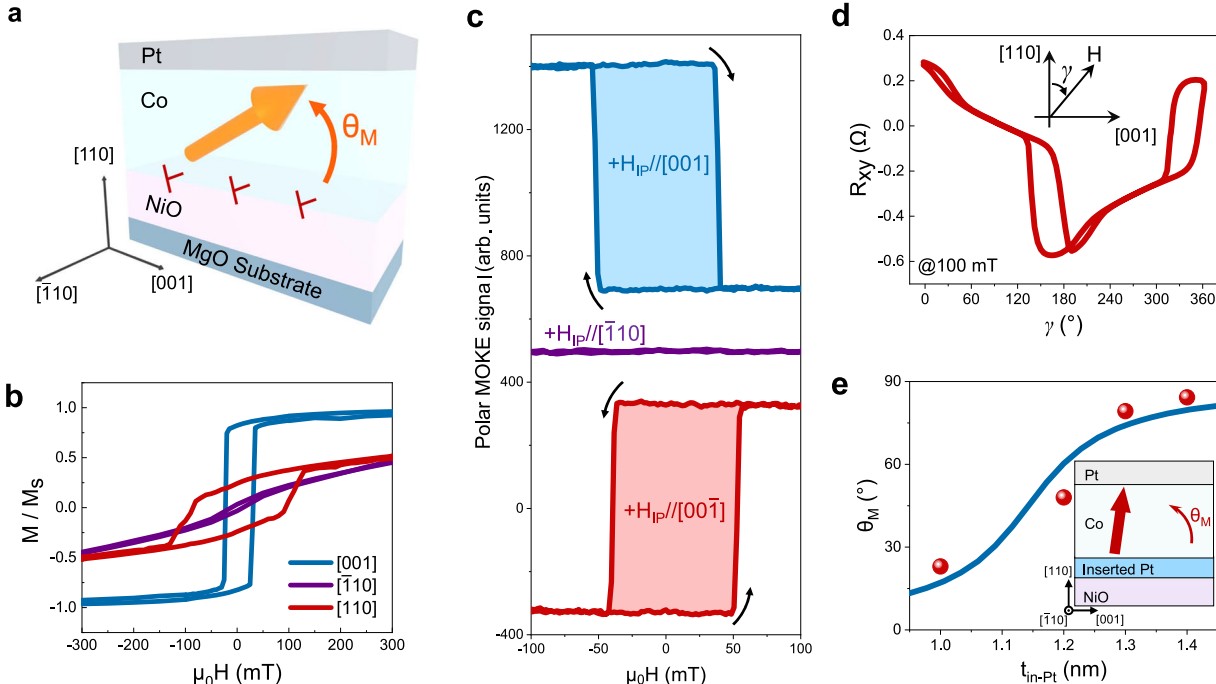

**Fig. 3 | The controllable tilted magnetic easy axis induced by in-plane symmetry breaking. a** The schematic diagram for the tilted magnetic easy axis induced by dislocations in (110)-orientation Pt/Co/NiO/MgO heterostructures. **b** The magnetic hysteresis loops along the [001]-, [$\bar{1}$10]-, [110]-direction of (110)-orientation Pt(5)/Co(1.2)/NiO(20)/MgO. **c** Field dependence of polar MOKE signals by sweeping the magnetic field along in-plane [001] and [$\bar{1}$10] directions. Note that polar MOKE signals mainly correspond to the perpendicular magnetization. A data shift is induced for better visualization. **d** The polar angular ($\gamma$) dependent Hall resistance ($R_{xy}$) by rotating external magnetic field H in the ($\bar{1}$10) plane. The inset is the geometry set for the measurement. **e** The modulation of $\theta_M$ by increasing the thickness of inserting Pt layer ($t_{in\text{-}Pt}$), with blue line as guideline to the eye. The inset is the schematic diagram of the sample for the controllable tilted magnetic easy axis by inserting Pt layer between the Co and NiO layers.

The observation of tilted magnetic easy axis in the (110)-orientation heterostructures implies its strong correlation to the tilting of crystal lattice. Below we discuss the underlying mechanisms by considering different contributions to the magnetic anisotropy of the Pt/Co heterostructures, including shape anisotropy, magneto-crystalline anisotropy, magneto-elastic anisotropy and interfacial anisotropy. The shape anisotropy of ultrathin films favors the magnetic easy axis within the film plane. As revealed by previous reports, the presence of easy (hard) axis along [001] ([$\bar{1}$10]) in-plane direction is mainly determined by the magneto-elastic effect[36,37]. Moreover, the interfacial contribution leads to an easy axis along the film normal direction[29,38]. By considering the energy competition using a phenomenological model, we can determine the angular dependence of the anisotropy energy and thereby the direction of magnetic easy axis (see Supplementary Note 3 and Supplementary Fig. 16). When the crystal lattice of (110)-orientation Co/Pt does not show any tilting ($\theta_e = 0°$), our simulation results show that the magnetic easy axis would be along either the out-of-plane [110] ($\theta_M = 90°$) or in-plane [001] ($\theta_M = 0°$) axis. Moreover, we find that the presence of crystal lattice tilting ($\theta_e > 0°$) affects the angular dependence of anisotropy, leading to the tilting of magnetic easy axis within the ($\bar{1}$10) plane. Our simulation results further reveal that the magnitude of $\theta_M$ depends on both the tilting angle of crystal lattice ($\theta_e$) and the relative strength of anisotropy energies. As shown in Fig. 3e, we find that the $\theta_M$ can be continuously tuned from 16° to 84° by inserting a thin Pt layer with controlled thicknesses ($t_{in\text{-}Pt}$) between Co and NiO (see details in Supplementary Fig. 17), which is most likely due to the modulation of interfacial perpendicular magnetic anisotropy energy[37] and agrees well with our simulation results.

## Field-free SOT switching of perpendicular magnetization

We finally study the field-free SOT switching in the (110)-orientation heterostructures. Two representative types of heterostructures are used for demonstration, with different magnitudes of $\theta_M$ as schematically shown in Fig. 4a ($\theta_M = 16°$) and Fig. 4b ($\theta_M = 84°$). Current pulses ($I_{pulse}$) are injected along the Hall bar channels (see Supplementary Fig. 18), which are along both in-plane [$\bar{1}$10] and [001] directions. Perpendicular magnetization is probed by measuring the anomalous Hall resistance $R_{xy}$. Figure 4c, d show the changes of Hall resistance ($\Delta R_{xy}$) by sweeping $I_{pulse}$ without magnetic field for the two heterostructures respectively. When $I_{pulse}$ is along [$\bar{1}$10], the deterministic switching of perpendicular magnetization is observed for both heterostructures. Compared with remnant Hall resistance at $\mu_0 H = 0$ mT, the field-free SOT switching ratio is estimated to be about 70% for the $\theta_M = 16°$ sample, and about 60% for the $\theta_M = 84°$ sample (see Supplementary Fig. 19). By contrast, no obvious switching events occur when $I_{pulse}$ is along [001] direction in both heterostructures, see Fig. 4c, d. The SOT efficiencies as current applied along both in-plane directions are also estimated by second Harmonic Hall resistance[14,39,40], showing values close to similar heterostructures[41] (see Supplementary Note 4 and Supplementary Fig. 20). Further measurements by varying the magnitude and pulse-width of currents indicate that the Joule heating effect is likely to assist the switching of perpendicular magnetization in our samples (see Supplementary Note 5, Supplementary Figs. 21 and 22). Nevertheless, the distinct switching behaviors as current pulses applied along two in-plane directions in (110)-orientation heterostructures, along with the absence of field-free SOT switching in (100)- and (111)-orientation heterostructures (see Supplementary Fig. 23),

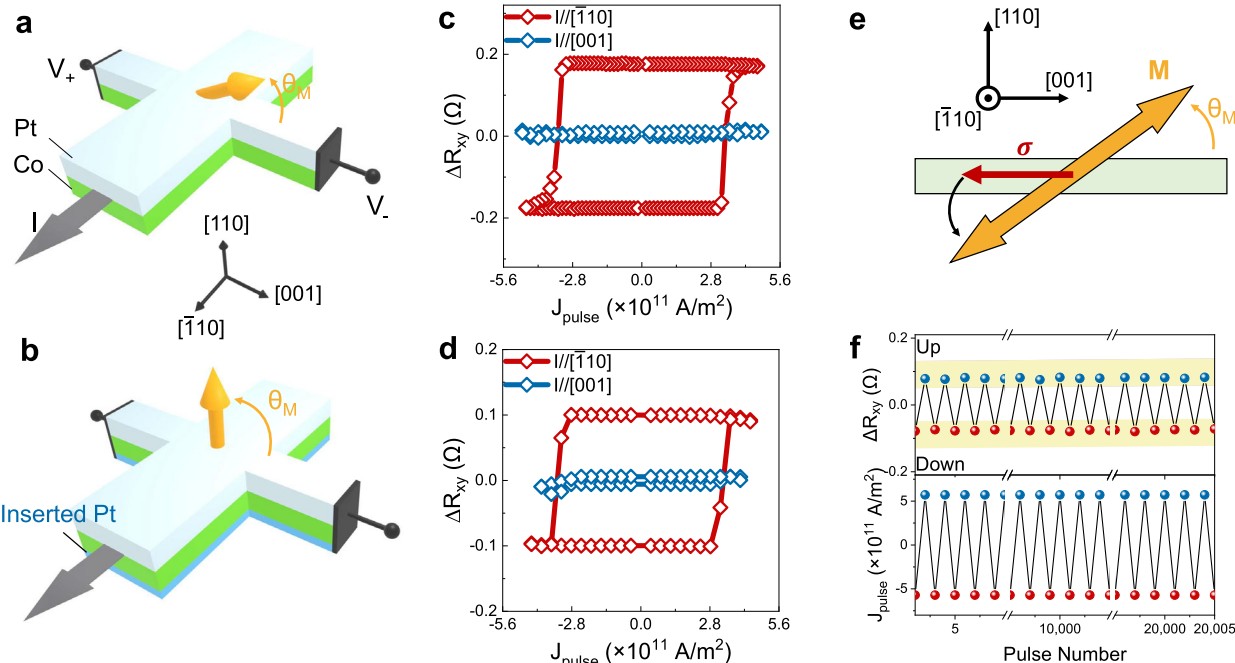

**Fig. 4 | Field-free current-induced switching of magnetization with perpendicular components.** The schematic diagram of the set-up for SOT switching of perpendicular magnetization is shown in **a**, with small $\theta_M$ and **b**, with large $\theta_M$. The yellow arrows indicate the ferromagnetic magnetization. The black arrows indicate the $I_{pulse}$. The results of field-free SOT switching with $I_{pulse}$ (pulse width of 2 ms) applied along [$\bar{1}$10] and [001] are shown in **c**, for Pt(5)/Co(1.2)/NiO(20)/MgO heterostructure with $\theta_M = 16°$ and **d**, for Pt(5)/Co(0.6)/Pt(1.4)/NiO(20)/MgO heterostructure with $\theta_M = 84°$, respectively. Field-free SOT switching is only observed as

$I_{pulse}$ is applied along [$\bar{1}$10]. **e** The schematic diagrams illustrate the switching of magnetization by SOT. The yellow arrow indicates the magnetization. The red arrow indicates the spin polarization $\sigma$ provided by Pt as $I_{pulse}$ is applied along [$\bar{1}$10] direction. The black curve arrow indicates the direction of magnetization relaxation. **f** Reversible field-free SOT switching with $\pm J_{pulse}$ about $5.7 \times 10^{11}$ A/m$^2$ and pulse width of 300 μs along [$\bar{1}$10] in (110)-orientation Pt(5)/Co(0.6)/Pt(1.4)/NiO(20)/MgO heterostructure for more than $10^4$ cycles. Values in parentheses are thickness in nm.

demonstrate the essential role of tilted anisotropy induced by the tilting of crystal lattice.

These results above can be explained by considering the symmetry between spin polarization ($\sigma$) induced by the spin-Hall effect in Pt and the tilted magnetic easy axis. As schematically shown in Fig. 4e, by applying current along [$\bar{1}$10] axis, the $\sigma$ provided by Pt exerts a damping-like torque $\tau_{DL}$ on the magnetization of Co and forces the magnetization to align parallel to $\sigma$ during $I_{pulse}$[7,10,18]. After removing the $I_{pulse}$, the magnetization will relax to downward direction due to the presence of the tilted magnetic easy axis. Similarly, inverting current direction switches the magnetization from downward to upward, leading to the deterministic switching behaviors. This scenario is further supported by the field-dependent SOT switching when external magnetic fields are applied parallel to the current pulse, see Supplementary Note 6, Supplementary Figs. 24 and 25. The critical switching current density $J_c$ of our device is about $3 \times 10^{11}$ A/m$^2$ (normalized by thickness and width of Hall bar), which is comparable or better than the other typical field-free SOT systems at room temperature using the conventional ferromagnetic and heavy metals, such as CoFeB/Ta/CoFeB with interlayer coupling[10] (~$10^{12}$ A/m$^2$) and Pt/Co/NiO with exchange bias[13] (~$10^{12}$ A/m$^2$). Moreover, we also test the stability and endurance of field-free switching events by reversibly applying current pulses along [$\bar{1}$10] direction for more than $10^4$ cycles. As shown in Fig. 4f and Supplementary Fig. 26, our results reveal that the field-free switching is highly reversible without clear fatigue.

As the in-plane symmetry breaking is enabled by the design of dislocations, our strategy to realize the field-free SOT switching should be potentially extended to other materials. Indeed, we also successfully realize the field-free SOT switching of perpendicular magnetization in Pd-based heterostructures on (110)-orientation oxide

templates, see Supplementary Fig. 27. Moreover, compared with other approaches to realize tilted magnetic anisotropy[17,42,43], our strategy does not rely on complex fabrication process or delicate interlayer coupling, and exhibits excellent performances (see Supplementary Note 7 and Supplementary Table 1). Given that oxides with different orientations can be deposited on silicon[44], we believe this strategy to break in-plane crystal symmetry by designing the orientation of Burgers vectors **B** provides a simple and applicable path approaching the field-free SOT devices.

In conclusion, we have demonstrated that the in-plane symmetry in magnetic heterostructures of benchmark SOT materials can be broken by properly designing the orientation of Burgers vector **B** of dislocations. Such an in-plane symmetry breaking leads to a tilted magnetic easy axis that can be tuned from being nearly in-plane to nearly out-of-plane, and enables a reliable field-free SOT switching of perpendicular magnetization at room temperature with relatively low current density and excellent stability. This revelation has a broader implication for designing the symmetry of functional materials, which include but not limit to magnetoelectronics, ferroelectrics, topotronics, antiferromagnetic spintronics, etc.

## Methods
### Sample fabrication
Conventional pulse layer deposition (PLD) method was utilized to fabricate an epitaxial NiO layer, using a KrF excimer pulsed laser with a wavelength of 248 nm, repetition rate of 5 Hz and energy density as 1.4 J/cm$^2$. The deposition temperature and oxygen pressure were controlled at 650 °C and 6.7 Pa, respectively. The distance between target and substrate was set as 8 cm, to obtain a stable deposition rate of 0.8 nm/min. The miscut angle of MgO substrates is less than 0.2°.

After deposition, the NiO film was immediately transferred to a magnetron sputtering chamber with a background vacuum around $2 \times 10^{-8}$ Torr, heated to 150 °C for 15 min, to avoid possible gas absorbed on the surface. Then dc sputtering was employed to deposit Co and Pt layers at room temperature. During the deposition, the substrate was rotated to ensure spatial homogeneity. The Co layer was deposited with power as 15 W under 3 mTorr, and the rate was controlled as 0.6 nm/min. The Pt layer was deposited with power as 30 W under 2 mTorr, and the rate was controlled as 2 nm/min. The samples were patterned into Hall bar devices with a width of 20 μm by photolithography and Ar ion etching for further transport measurement and current-switching experiments. Electrodes and contact pads were formed by using photolithography and a lift-off process.

### HAADF-STEM experiments and the determination of tilting of crystal lattice

Cross-sectional STEM samples were prepared using focused ion beam (FIB, Zeiss Auriga) with $Ga^+$ ions. Samples were thinned to less than 100 nm at an accelerating voltage of 30 kV with a decreasing current from the maximum of 600 pA, followed by fine polishing at an accelerating voltage of 5 kV with a small current of 20 pA, and then further milled with Ar ions to remove the surface damage. The STEM experiments were performed on an electron microscope equipped with double aberration correctors (Titan Cubed Themis G2 300). The atomic-resolved HAADF images were collected with a convergence angle of 25 mrad and a collection angle of 64–200 mrad. To determine the tilting of crystal lattice on the metal/oxide interface, we determined the directions of adjacent atoms arrows of metal layer and NiO near metal/oxide interface, respectively, and the angle between the two directions is defined as tilting angle $\theta_e$.

### Characterization

The epitaxy relationship to the substrate was characterized by Empyrean high-resolution X-ray diffractometer produced by Company Panalytical with Cu $K_{\alpha1}$. A Superconducting quantum interfere device (SQUID) magnetometer was utilized to characterize basic magnetic properties of samples by vibrating magnetometer mode. The polar MOKE microscopy produced by Evico (German) was employed to measure the change of polar MOKE signal with the application of in-plane magnetic field. The synchrotron XRD data were obtained at 1W1A Diffuse X-ray Scattering Station, Beijing Synchrotron Radiation Facility (BSRF-1W1A). The transport property was measured in Physical Property Measurement System (PPMS). In current-induced SOT switching of perpendicular magnetization experiments, a pulsed current with a duration of 2 ms was applied for each data point. After 6 s, a small a.c. excitation current (0.5 mA) is utilized to measure the Hall resistance.

## Data availability

The data supporting the findings of this study are available within the paper and its Supplementary Information and Supplementary Data, and also from the corresponding author upon reasonable request. Source data are provided in this paper.

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

## Acknowledgements

We thank Prof. Nian-Xiang Sun and Dr. Liang Wu for fruitful discussions. We thank Dr. Hao Bai and Dr. Yiqing Dong for SOT experiments. A portion of this work is based on the data obtained at BSRF-1W1A. The authors gratefully acknowledge the cooperation of the beamline scientists at BSRF-1W1A beamline. Y.-H. L and R. Y. were supported by the Basic Science Center Project of the National Natural Science Foundation of China (NSFC) (grant number 52388201). W.-J.J. thanks the National Natural Science Foundation of China (Grant Nos. 52271181, 51831005), the National Key R&D Program of China (Grant No. 2022YFA1405100), the NSF Distinguished Young Scholars Grant (No.12225409), Beijing Natural Science Foundation (Grant No. Z190009), the Beijing Advanced Innovation Center for Future Chip (ICFC). D. Y. was supported by National Natural Science Foundation of China (52002204, 92163113). T. N. was supported by the National Natural Science Foundation of China (52073158, 52161135103) and the National Key R&D Program of China (2021YFA0716503). Y. Z. was supported by the Natural Science Foundation of China (Grant No. 52002370). J.-M. H. acknowledges support from the NSF award CBET-2006028. J. S. is grateful for the partial financial support by JSPS KAKENHI Grant Number 19H00864.

## Author contributions

Y.-H.L., Y.L., D.Y., and T.N. conceived this study. Y.L., D.Y., and T.N. performed this study under the supervision of Y.-H. L. J.-M.H., Y.L., H.C., and M.D. performed the phenomenological simulations. Y.L., D.Y., and T.N. fabricated the samples and carried out the transport and magnetic measurements. R.Y. and S.L. performed the STEM characterizations. W.-J.J., Y.L., and T.X. conducted the SOT measurement. W.-J.J., Y.L., and L.Z. conducted the polar MOKE characterization. J.S., B.X., Y.L., and Y.-H.L. discussed and analysis the results. Y.L., D.Y., T.N., W.-J.J., R.Y., and Y.-H.L. wrote the manuscript. All authors discussed the results and revised the manuscript.

## Competing interests

The authors declare no competing interests.
