## [Peer Review File · Nature Communications]

Reviewers' Comments:

Reviewer #1:

Remarks to the Author:

Comments on the manuscript "Field-free Spin-orbit Switching of Perpendicular Magnetization Enabled by Dislocation-induced In-plane Symmetry Breaking"

All-electrical manipulation of perpendicular magnetization has long been an interesting topic for the last decade. Many groups have investigated the Pt/Co heterostructures as a typical PMA system. In this manuscript, it was demonstrated possible to design the orientation of the Burgers vectors of dislocations using different MgO substrates with various orientations. Furthermore, the proper Burgers vector on MgO (110) introduced a tilted PMA easy axis for the Pt/Co system, which assisted the desired field-free SOT switching for the system. This work is physically interesting and may be influential in providing a new proposal to realize practical SOT devices. The layout of the manuscript is also logical and clear. I think the manuscript can be given a further consideration after addressing the following issues.

1. To make the Burgers vectors with an out-of-plane component a useful mechanism to introduce the tilted PMA, the authors should not only evaluate the tilting angle θ_e of crystalline planes near the dislocation (shown in Supplementary Figure S3 and S4) but also evaluate the density of the dislocations and make sure the mechanism work for the whole stack. In this current version, the HAADF images have showed a zoomed-in area. The wide-range distribution of the desired dislocations should be characterized or estimated, which is important for practical applications.
2. The authors mentioned the structure have application potentials. For applicable SOT memory devices, magnetic tunnel junctions fabricated on Si/SiO₂ substrate are needed currently. A discussion how to integrate this dislocation mechanism with the sophisticated MRAM technology is helpful.
3. In Fig.4(d), why is there a partial switching loop for the blue line as $I//[001]$ compared with 4(c)?
4. The experiments were conducted at room temperature; how about the influence of the heating effect in the SOT switching measurement? What is the switching degree by SOT relative to the R_{xy} in the R_{xy} -Hoop curve?
5. How about the damping-like and field-like torque efficiency for the system and compare the measured value with other references?
6. To illustrate the influence of the Burgers vectors with an out-of-plane component (the stack grown on (110) MgO), the other samples grown on the (111) and (001) MgO substrates can be ideal control samples. Using these control samples, other entangling factors involved if any during the film or device fabrication process can be clearly ruled out. I suggest the author to conduct the same tilting angle measurement and the SOT-switching measurement for the (111) and (001) samples as well. And further check whether (1) the easy axis tilting is absent and (2) field-free switching is not achievable for the control device.
7. The switching performance under different fields for example, (-1000 Oe, +1000 Oe) with a spacing of 100 Oe or even smaller can be given. In this case, readers can estimate an effective in-plane field introduced by the tilted easy axis by observing the critical field condition at which SOT switching is disabled.
8. It also seems strange why a small tilting angle of about 1° in the atomic structure can introduce such a large tilting PMA angle as 30° (tilted by 60° as shown in Figure 3a&e). So the characterization of the tilting angle is still puzzling for me. In many cases, we also observed a sharp switching in the R_{xy} vs H_{in} -plane curves similar with Figure 3c. However, this curve cannot guarantee a tilted PMA. I suggest the authors conduct the sample measurement of Figure 3c for the control devices on (111) and (100) substrates and check the perpendicular magnetization switching by the in-plane field is still possible or not?
9. Regarding the critical current density, it is strongly dependent on the pulse duration. How about the current pulse duration used in this experiment? If the pulse is as long as above microsecond, the current density of $3.0 \times 10^7 \text{ A/cm}^2$ is a moderate value.
10. There is a typo in Page 5. "tiling" \diamond "tilting".
11. Is there any typo for the current density of $\sim 10^{15} \text{ A/m}^2$ for the Pt/Co/AlO_x system? This value seems not physical.
12. In the legend of Fig.4d "for Pt(5)/Co(0.6)/Pt(1.4)/NiO(20)/MgO heterostructure with $\theta_M \sim 90^\circ$ ",

I suggest a more concise value is used instead of 90°. If the angle is 90°, field-free switching is physically impossible. A more concise value smaller than 90° would appear more realistic and convincing and not misleading.

Reviewer #2:

Remarks to the Author:

The authors proposed a new strategy of field-free spin-orbit torque (SOT) switching by introducing in-plane symmetry breaking using dislocations with an out-of-plane Burgers vector. The out-of-plane Burgers vector induces a tilted magnetic easy axis that enables field-free SOT switching of perpendicular magnetization. The authors claimed that their method is superior to previously demonstrated field-free SOT switching ones as it can be generally applicable to various material systems. However, I disagree with this argument for the following reasons.

First, to design the orientation of Burgers vectors of dislocations, they use a single crystalline MgO substrate and a NiO underlayer, on which Pt grows hetero-epitaxially enabling control of dislocations. I wonder if this can be applied to other SOT materials such as W or Ta. Can the authors demonstrate the applicability of this method to other material systems, either experimentally or through calculation? Moreover, I suggest that the authors should discuss the merits of their method compared to already reported ones showing field-free SOT switching using tilted magnetic anisotropy [Zhi Li, ACS Appl. Electron. Mater. 2022, H. Kim, Adv. Func. Mater. 2022, L. You, PNAS 2015].

Secondly, I wonder if the magnetization is fully switched by SOT without a magnetic field. It seems odd to me that the change in Hall resistance (ΔR_{xy}) for the sample with $\theta_M=16$ degrees is much larger than that for the sample with $\theta_M \sim 90$ degrees. The authors should compare the field-free SOT switching results to the anomalous Hall resistance measured with perpendicular magnetic fields and show how much the magnetization is switched. I don't think it is reasonable to compare critical switching current density with other results unless full switching is achieved.

Based on the above, I don't find significant advances in this manuscript to be published in Nature Communications. Therefore, I cannot recommend this manuscript.

Reviewer #1:

All-electrical manipulation of perpendicular magnetization has long been an interesting topic for the last decade. Many groups have investigated the Pt/Co heterostructures as a typical PMA system. In this manuscript, it was demonstrated possible to design the orientation of the Burgers vectors of dislocations using different MgO substrates with various orientations. Furthermore, the proper Burgers vector on MgO (110) introduced a tilted PMA easy axis for the Pt/Co system, which assisted the desired field-free SOT switching for the system. This work is physically interesting and may be influential in providing a new proposal to realize practical SOT devices. The layout of the manuscript is also logical and clear. I think the manuscript can be given a further consideration after addressing the following issues.

Response: We thank this referee's for recognizing our work being "physically interesting and may be influential in providing a new proposal to realize practical SOT devices". Following her/his insightful suggestions, we have now performed additional experiments and analyses in this revised manuscript and supplementary materials. Below we present our responses to specific comments.

1. To make the Burgers vectors with an out-of-plane component a useful mechanism to introduce the tilted PMA, the authors should not only evaluate the tilting angle θ_e of crystalline planes near the dislocation (shown in Supplementary Figure S3 and S4) but also evaluate the density of the dislocations and make sure the mechanism work for the whole stack. In this current version, the HAADF images have showed a zoomed-in area. The wide-range distribution of the desired dislocations should be characterized or estimated, which is important for practical applications.

Response:

We thank the referee for making this insightful comment. The dislocation arrays in (100)-oriented and (110)-oriented heterostructures are confirmed by geometry phase analysis, as shown in **Figure R1** below. For (100)-oriented and (110)-oriented heterostructures, the density of dislocation is estimated to be about $4.11 \times 10^6/\text{cm}$, and $3.70 \times 10^6/\text{cm}$, respectively. Considering the lattice constant of Pt (3.9 Å), Co(FCC) (3.55 Å), the theory model predicts the density of dislocation to be about $1.44 \sim 3.57 \times 10^6/\text{cm}$, which is very close to our experimental results. In addition, we also utilize synchrotron based XRD to measure the tilting of Pt lattice, which is also confirmed as shown in Fig. 7 of Supplementary Materials. Thus the presences of lattice tilting induced by dislocation are identified in the whole samples, rather than selected areas. This issue is mentioned on page 5 of revised manuscript. These

results are now presented in Fig. 6 and Fig. 7 of Supplementary Materials.

Figure R1. The dislocation arrays confirmed by geometry phase analysis (GPA). **a**, the HADDF image and GPA analysis result of Pt(5 nm)/Co(1.5 nm)/NiO(20 nm)/MgO(110). **b**, the HADDF image and GPA analysis result of Pt(5 nm)/Co(1.5 nm)/NiO(20 nm)/MgO(100). The dislocations are denoted by white arrows.

2. The authors mentioned the structure have application potentials. For applicable SOT memory devices, magnetic tunnel junctions fabricated on Si/SiO₂ substrate are needed currently. A discussion how to integrate this dislocation mechanism with the sophisticated MRAM technology is helpful.

Response:

We acknowledge this important suggestion. According to *J. Appl. Phys.* **91**, 5728-5734 (2002), the highly textured (110)-oriented MgO film can be deposited on (100)-Si substrate by PLD. Thus, it is possible to fabricate (110)-oriented heterostructures on (100)-Si substrate by combining PLD and magnetron sputtering techniques. Thereby, we believe it is possible to integrate the dislocation mechanism on the silicon-based MRAM technology. This part has been discussed on page 10 of the revised manuscript.

3. In Fig.4(d), why is there a partial switching loop for the blue line as $I//[001]$ compared with 4(c)?

Response:

We thank this referee for noting this issue. It is noted that the variation of R_{xy} of the blue line is very tiny (about 3% of magnetic switching) as compared with the other direction. To further clarify this issue, we also conduct the SOT switching experiments again and change the duration of pulse (from 2 ms to 500 us), see **Figure R2** below. It is likely that the change of R_{xy} as $I//[001]$ can be ascribed as artifact that were introduced by the thermal effect. To avoid ambiguity, we also incorporate these results in Fig. 20 of Supplementary Materials.

Figure R2. The artifact introduced by thermal effect in SOT switching as current pulse along $[001]$ direction of (110) -oriented heterostructure with $\theta_M = 84^\circ$.

4. The experiments were conducted at room temperature; how about the influence of the heating effect in the SOT switching measurement?

Response:

We appreciate this important suggestion. Considering the Joule heating effect ($\propto I^2R$), the device temperature should be raised up during the application of current pulses. Here, we check the temperature increase of the representative devices. We first measure the longitude resistance (R_{xx}) in devices with $\theta_M = 16^\circ$ and $\theta_M = 84^\circ$, which is done by using a small dc current of 0.1 mA, see **Figure R3a**. The dc current of 0.1 mA is too small to induce an appreciable Joule heating. Then we measure the R_{xx} as a function of the amplitude of the ac current density ($R_{xx} \sim J$), see **Figure R3b**. The temperature raises during SOT switching can be estimated by comparing the temperature-induced resistance changes with the

current-induced resistance changes. The current-induced temperature increase (ΔT) in SOT switching process can reach up to about 95 K for $\theta_M = 16^\circ$ device and about 80 K for $\theta_M = 84^\circ$ device. Note that the temperature rising in our devices is likely to be overestimated by this method, considering that the pulse width for SOT switching is only 2 ms. In addition, we measure the temperature dependent AHE of the aforementioned devices (300 K to 380 K), see **Figure R3c**. As the temperature increases, the magnetization slightly tilts towards the film plane and coercivity decreases, which could be beneficial for the field-free SOT switching. Thus, the Joule heating could play a partial role in the current-induced field-free SOT switching, in the presence of the dislocation-induced in-plane symmetry breaking. We also mention this issue on page 9 of manuscript. The results are included in Note 5 and Fig. 19 of Supplementary Materials.

Figure R3. The estimated Joule heating induced by current pulses. **a**, the temperature dependent R_{xx} with *dc* current of 0.1 mA. **b**, estimated temperature increase by monitoring the R_{xx} under various *ac* current. The red circles represent the experimental data, and the blue lines represent the parabolic fitting curve. **c**, the temperature-dependent AHE resistance. Upper panel corresponds to heterostructure with $\theta_M = 16^\circ$ and lower panel corresponds to heterostructure with $\theta_M = 84^\circ$.

5. What is the switching degree by SOT relative to the R_{xy} in the R_{xy} -Hoop curve?

Response:

For device with $\theta_M = 16^\circ$, the field-free switching ratio is about 70%, see **Figure R4** below. For devices with $\theta_M = 84^\circ$, measurements on multiple devices reveal that the field-free switching ratio ranges from 40% to about 60%, see **Figure R5** below. Note that these

switching ratios are similar to those reported results in *Nat. Nanotechnol.* **14**, 939-944 (2019) (switching ratio about 20%), and *Nat. Mater.* **20**, 800-804 (2021) (switching ratio about 70%). The incomplete field-free switching can be attributed to the pinning effect and current shunting from Hall voltage arms. For devices with switching ratio about 60%, the endurance test also confirms a good stability, as shown in Figure R6 below. This issue is mentioned on page 9 of revised manuscript. The results are included in Fig. 17 of Supplementary Materials.

Figure R4. The SOT switching ratio in sample with $\theta_M = 16^\circ$. The dash lines denote the remnant R_{xy} in the zero field.

Figure R5. The device-variation of field-free SOT switching in sample with $\theta_M = 84^\circ$. The current pulse is applied along $[\bar{1}10]$.

Figure R6. Endurance test of field-free SOT switching with $\pm J_{\text{pulse}}$ of 5.7×10^{11} A/m² along $[\bar{1}10]$, which consistently confirms the switching ratio about 60%. **a**, the pulse cycle for more than 10^4 cycles. **b**, the field-free SOT switching loop before and after conducting the endurance test. The pulse width is 300 μ s.

6. How about the damping-like and field-like torque efficiency for the system and compare the measured value with other references?

Response:

Following reviewer's suggestion, both the field-like and damping-like torque efficiencies were measured, which are also compared with other reference. We utilized second harmonic Hall (SHH) resistance to measure the spin-torque efficiency. The measurement geometry is shown in **Figure R7a** below, the *ac* current $I = I_0 \sin(\omega t)$ is applied along the x-direction (x-direction along $[\bar{1}10]$ or $[001]$). Two lock-in amplifiers are utilized to record the first ($V_{1\omega}$) and second ($V_{2\omega}$) harmonic Hall voltage during sweeping in-plane magnetic field H_x . The first harmonic Hall voltage $V_{xy}^{1\omega}$ mainly corresponds to the direction of magnetization. The second harmonic Hall voltage $V_{xy}^{2\omega}$ is correlated to the oscillation of magnetization around the equilibrium position, which is induced by SOT-generated effective field.

As the in-plane magnetic field H_x larger than anisotropic field H_K , the magnetization vector \mathbf{M} is aligned along the x-direction, leading to the SHH resistance to be written as [*Phys. Rev. Lett.* **123**, 207205 (2019); *Nat. Commun.* **12**, 4555 (2021)]:

$$R_{xy}^{2\omega} = \frac{R_{\text{AHE}}}{2} \frac{H_{\text{DL}}}{|H_x| - H_K} + R_{\text{PHE}} \frac{H_{\text{FL}}}{|H_x|} + R_{\text{thermal}}$$

Where R_{AHE} and R_{PHE} are the AHE and planar Hall resistances and R_{thermal} is the thermal contribution from anomalous Nernst and spin Seebeck effects. Since $R_{\text{PHE}} \ll R_{\text{AHE}}$, the second term can be neglected. Thus, we can estimate the effective field H_{DL} via fitting the $R_{\text{xy}}^{2\omega}$ data in the large in-plane field regime.

We then apply *ac* current along [001] and $[\bar{1}10]$ directions with in-plane field parallel to the current in sample Pt(5)/Co(0.6)/Pt(1.4)/NiO(20)/MgO(110), these results are shown in **Figure R7b** and **R7c** below. By changing the amplitude of *ac* current and fitting the SHH resistance, as shown in **Figure R7d** below. The H_{DL}/J_e along [001] is estimated to be 4.25 mT per 10^{11} A/m², and the H_{DL}/J_e along $[\bar{1}10]$ is about 4.83 mT per 10^{11} A/m². These values are close to the recorded value in similar Pt/Co heterostructures (*Phys. Rev. B* **93**, 144409 (2016)). In addition, the damping-like spin-torque efficiency can be estimated as $\xi_{\text{DL}} = \frac{2e}{h} M_s t_{\text{Co}} \frac{H_{\text{DL}}}{J_e}$. Here, we take the saturated magnetization of Co as $M_s = 1000$ emu/cm³, and $t_{\text{Co}} = 0.6$ nm for the thickness of Co. By taking the H_{DL}/J_e of two directions, the damping-like torque efficiency ξ_{DL} along $[\bar{1}10]$ ($\xi_{\text{DL}}^{[\bar{1}10]}$) is calculated as 0.088, and ξ_{DL} along [001] ($\xi_{\text{DL}}^{[001]}$) is about 0.078, respectively.

We also apply transverse fields in SHH resistance measurement, e.g. current along $[\bar{1}10]$ ([001]) as field is along [001] ($[\bar{1}10]$), which can characterize the field-like effective field (*Phys. Rev. Appl.* **11**, 034018 (2019)). However, the $R_{\text{xy}}^{2\omega}$ measured by transverse magnetic field is about 1 or 2 order lower than the $R_{\text{xy}}^{2\omega}$ measured by longitude magnetic field (see the insets of **Figure R7b** and **R7c**). Thus, the field-like torque can be neglected and the damping-like torque plays the main role in SOT switching, which is consistent with previous studies (*Nat. Commun.* **12**, 4555 (2021); *Phys. Rev. Lett.* **109**, 096602 (2012)). This issue is mentioned on page 9 of revised manuscript. These results are discussed and included in Note 4 and Fig. 18 of Supplementary Materials.

Figure R7. SOT efficiency measured by harmonic Hall measurements. **a**, the schematic diagram of the measurement geometry. The magnetization \mathbf{M} is oscillated by H_{DL} . The yellow arrows indicate the direction of magnetization \mathbf{M} . The experimental results of $R_{xy}^{2\omega}$ are shown in **b**, ac current along $[\bar{1}10]$, and **c**, ac current along $[001]$. The blue circles are the experimental data, and the red lines are the fitting curves. The insets are the $R_{xy}^{2\omega}$ with external field H along $[001]$ or $[\bar{1}10]$, applying ac current with same magnitude. **d**, the effective field H_{DL} as a function of current density. The red and blue lines represent the linear fitting.

7. To illustrate the influence of the Burgers vectors with an out-of-plane component (the stack grown on (110) MgO), the other samples grown on the (111) and (001) MgO substrates can be ideal control samples. Using these control samples, other entangling factors involved if any during the film or device fabrication process can be clearly ruled out. I suggest the author to conduct the same tilting angle measurement and the SOT-switching measurement for the (111) and (001) samples as well. And further check whether (1) the easy axis tilting is absent and (2) field-free switching is not achievable for the control device.

Response:

We thank this referee for making this constructive comment. Following her/his suggestion, we have now characterized the magnetic anisotropy of (100) - and (111) -oriented

heterostructures and its relation with the SOT switching ratio. We utilize the polar angular (γ) dependent R_{xy} measurement as shown in Fig. 3d. See **Figure R8** below, the polar angular (γ) dependent R_{xy} shows that the tilted magnetic easy axis is absent in (100)- and (111)-oriented heterostructures, which is also consistent with magnetometer results in Fig. 13 of Supplementary Materials. We then conduct the SOT switching experiments in (100)- and (111)-oriented heterostructures, as shown in **Figure R9** below. The field-free SOT switching is also absent. Note that this comment is related to Question 9 below. This issue is mentioned on page 9 of revised manuscript. These results are incorporated into Fig. 13 and Fig. 21 of Supplementary Materials.

Figure R8. The polar angular (γ) dependent R_{xy} in (100)-oriented (111)-oriented heterostructures.

Figure R9. The absence of field-free SOT switching of perpendicular magnetization in (100)- and (111)-oriented heterostructures. The results are shifted for better visualization.

8. The switching performance under different fields for example, (-1000 Oe, +1000 Oe) with a spacing of 100 Oe or even smaller can be given. In this case, readers can estimate an effective in-plane field introduced by the tilted easy axis by observing the critical field condition at which SOT switching is disabled.

Response:

Stimulated by this insightful suggestion, we have now characterized the field-dependent SOT switching. The schematic of measurements is shown in **Figure R10** below, in which the external magnetic field is applied parallel to the current pulse. For Pt(5)/Co(1.2)/NiO(20)/MgO(110) ($\theta_M = 16^\circ$), as the current pulse is applied along $[\bar{1}10]$, the magnetic field up to 100 mT cannot suppress the SOT switching, see **Figure R10a** below. However, as the current pulse is applied along $[001]$, the current switching is prohibited regardless the choice of in-plane fields, as shown in **Figure R10b** below. For Pt(5)/Co(0.6)/Pt(1.4)/NiO(20)/MgO(110) ($\theta_M = 84^\circ$), as the current pulse is applied along $[\bar{1}10]$, the external magnetic field about 2.2 mT can compensate the SOT switching, see **Figure R10c** below. As the current pulse is applied along $[001]$, the SOT switching can be induced by in-plane magnetic field, see **Figure R10d** below.

Figure R10. The field-dependent SOT switching. The external magnetic field H is parallel to

current pulse. **a** and **b**, the field-dependent SOT switching in Pt(5)/Co(1.2)/NiO(20)/MgO(110) ($\theta_M=16^\circ$). **a**, the current pulse is applied along $[\bar{1}10]$. **b**, the current pulse is applied along $[001]$. **c** and **d**, the field-dependent SOT switching in Pt(5)/Co(0.6)/Pt(1.4)/NiO(20)/MgO(110) ($\theta_M=84^\circ$). **c**, the current pulse is applied along $[\bar{1}10]$. **d**, the current pulse is applied along $[001]$.

The distinct field-dependent SOT switching behavior of two heterostructures can be understood by considering the combination effects of the torque $\vec{\tau}_{ext}$ induced by external field \vec{H}_{ext} , SOT and the orientation of the easy axis (titled magnetic anisotropy). When the current pulse is applied along $[\bar{1}10]$, the damping-like torque forces the magnetization to align parallel to $\vec{\sigma}$ (along $[001]$ as shown in **Fig. R11a** and **R11c** below) (*Nat. Nanotechnol.* **14**, 939-944 (2019)). If the \vec{H}_{ext} is applied parallel to the current pulse, the $\vec{\tau}_{ext} \propto \vec{M} \times \vec{H}_{ext}$ will force the magnetization tilt towards the hard axis. For $\theta_M = 16^\circ$, the hard axis is far away from the sample-plane and the $\vec{\tau}_{ext}$ cannot align the magnetization towards the hard axis, thus the \vec{H}_{ext} up to 100 mT could not compensate the titled anisotropy, leading to a robust SOT switching behaviors, see **Figure R11a** below. For $\theta_M = 84^\circ$, the hard axis is close to the sample-plane ($\sim 6^\circ$), thus the \vec{H}_{ext} of 2.2 mT could compensate the titled anisotropy, leading to the change of SOT switching behaviors, see **Figure R11c** below.

When the current pulse is applied along $[001]$, as shown in **Figure R11b** and **R11d** below, the damping-like torque forces the magnetization to align parallel to $\vec{\sigma}$ (along $[\bar{1}10]$ as shown in **Fig. R11b** and **R11d**). It is noted that $[\bar{1}10]$ is the magnetic hard axis (see Fig. 2 of revised Manuscript) and the tilted easy axis is within the plane perpendicular to $[\bar{1}10]$. Therefore, deterministic SOT switching of perpendicular magnetization is not expected without \vec{H}_{ext} . For $\theta_M = 16^\circ$, the magnetic easy axis is very close to the $[001]$ -direction. Thus the \vec{H}_{ext} along $[001]$ would strongly favor magnetization along this direction, resulting in the absence of SOT switching, see **Figure R11b** below. For $\theta_M = 84^\circ$, the easy axis is close to out-of-plane $[110]$ -direction and the application of \vec{H}_{ext} would assist SOT switching (**Figure R11d**). This issue is mentioned on page 10 of revised manuscript, and these new results and discussions are now included in the Note 6 and Fig. 23 in Supplementary Materials.

Figure R11. The mechanism of field-dependent SOT switching. **a-b**, the effect of \vec{H}_{ext} in heterostructure with $\theta_M = 16^\circ$. **a**, the current pulse is applied along $[\bar{1}10]$, the hard axis is away from sample plane and the $\vec{\tau}_{ext}$ is insufficient to compensate SOT switching. **b**, the current pulse is applied along $[001]$, the magnetization is strongly favored by \vec{H}_{ext} , thus the SOT switching is absent. **c-d**, the effect of \vec{H}_{ext} in heterostructure with $\theta_M = 84^\circ$. **c**, the current pulse is applied along $[\bar{1}10]$, the $\vec{\tau}_{ext}$ could force the magnetization align along hard axis, and the SOT switching is compensated. **d**, the current pulse is applied along $[001]$, the $\vec{\tau}_{ext}$ strongly favors the magnetization and assists the SOT switching. The easy axis (E.A.) is denoted by brown lines, and hard axis (H.A.) is denoted by green dash lines.

9. It also seems strange why a small tilting angle of about 1° in the atomic structure can introduce such a large tilting PMA angle as 30° (tilted by 60° as shown in Figure 3a&e). So the characterization of the tilting angle is still puzzling for me. In many cases, we also observed a sharp switching in the R_{xy} vs $H_{in-plane}$ curves similar with Figure 3c. However, this curve cannot guarantee a tilted PMA. I suggest the authors conduct the sample measurement of Figure 3c for the control devices on (111) and (100) substrates and check the perpendicular magnetization switching by the in-plane field is still possible or not?

Response:

We appreciate this important suggestion. Firstly, as the reviewer suggested, we have characterized the switching of perpendicular magnetization by in-plane field in both (100)- and (111)-oriented heterostructures by using the same method in Manuscript Fig. 3c. As shown in **Figure R12**, no switching of polar MOKE signal is observed, which is consistent with previous analysis that suggests the tilted magnetic easy axis can only exist in (110)-oriented heterostructures. We also incorporate these new results into Supplementary

Fig.13 of the Supplementary Materials.

In addition, as discussed in Supplementary Note 3 and Supplementary Fig.14, the tilted anisotropy can be understood as the competition among perpendicular magnetic anisotropy, crystalline anisotropy and shape anisotropy. As shown in our simulation results in Supplementary Fig.14**b** of Supplementary Materials, when the effective perpendicular anisotropy energy is close to the crystalline anisotropy ($\xi = \frac{K_{\text{eff}}}{K_c}$ close to 1), a small tilting of crystalline lattice is enough to induced a tilted magnetic easy axis.

Figure R12. The absence of perpendicular magnetization switching in the presence of in-plane magnetic field for **a**, (100)-oriented heterostructure and **b**, (111)-oriented heterostructure.

10. Regarding the critical current density, it is strongly dependent on the pulse duration. How about the current pulse duration used in this experiment? If the pulse is as long as above microsecond, the current density of $3.0 \times 10^7 \text{ A/cm}^2$ is a moderate value.

Response:

We have now characterized the critical current density with different pulse durations. As shown in **Figure R13**, the critical switching current density of field-free SOT switching gradually increases as the pulsed widths reduce from 2 ms to 300 μs . This issue is mentioned on page 9 of revised manuscript. These results are also incorporated into Fig.20 of Supplementary Materials.

Figure R13. The pulse-width-dependent field-free SOT switching of perpendicular magnetization of (110)-oriented heterostructure with $\theta_M = 84^\circ$.

11. There is a typo in Page 5. “tiling” ”tilting”.

Response:

This typo has now been corrected. In addition, we have now performed a careful proof reading thorough this revised manuscript.

12. Is there any typo for the current density of $\sim 10^{15}$ A/m² for the Pt/Co/AlO_x system? This value seems not physical.

Response:

We are sorry for this confusion. This value is discussed in Supplementary Materials of *Science* **363**, 1435-1439 (2019), noted as ‘we find that the critical current density required to switch the OOP-IP element is around 4.7×10^{11} A/cm²’. However, as compared with previous studies in Pt/Co/AlO_x system, this value is relatively higher, and may be ascribed to a typo from the authors of *Science* **363**, 1435-1439 (2019). Therefore, we have deleted it in the revised manuscript.

13. In the legend of Fig.4d “for Pt(5)/Co(0.6)/Pt(1.4)/NiO(20)/MgO heterostructure with $\theta_M \sim 90^\circ$ ”, I suggest a more concise value is used instead of 90° . If the angle is 90° , field-free switching is physically impossible. A more concise value smaller than 90° would appear more realistic and convincing and not misleading.

Response:

We thank the reviewer for this suggestion. The easy axis is actually close to 84° as determined by anomalous Hall resistance, see Fig.15 of Supplementary Materials. In the initial manuscript, we used 90° to state that it is close to out-of-plane axis. We agree with the reviewer that this might cause confusion. Therefore, we have modified it as 84° .

Reviewer #2:

The authors proposed a new strategy of field-free spin-orbit torque (SOT) switching by introducing in-plane symmetry breaking using dislocations with an out-of-plane Burgers vector. The out-of-plane Burgers vector induces a tilted magnetic easy axis that enables field-free SOT switching of perpendicular magnetization. The authors claimed that their method is superior to previously demonstrated field-free SOT switching ones as it can be generally applicable to various material systems. However, I disagree with this argument for the following reasons.

Response:

We thank this referee for recognizing our strategy as “new” for the field of spin-orbitronics. In fact, it is our intention to design and demonstrate an effective method to introduce in-plane symmetry breaking that can facilitate the zero-field spin-orbitronic devices. Through incorporating the dislocations in the lattice mismatched epilayers, we have successfully demonstrated that the out-of-plane Burgers vector induces tilted magnetic anisotropy. Furthermore, we show that the resultant zero-field SOT switching occurs with an endurance up to 10^4 cycles. Meanwhile, stimulated by the insightful suggestions and comments made by both referees, we have extended necessary discussions in the main text and Supplementary Materials. We believe the clarity and robustness of this revised manuscript are largely enhanced. And we are hoping the referee is now in the position of supporting the publication of our manuscript.

1. First, to design the orientation of Burgers vectors of dislocations, they use a single crystalline MgO substrate and a NiO underlayer, on which Pt grows hetero-epitaxially enabling control of dislocations. I wonder if this can be applied to other SOT materials such as W or Ta. Can the authors demonstrate the applicability of this method to other material systems, either experimentally or through calculation?

Response:

We appreciate this insightful suggestion. To demonstrate the applicability of our method to other materials systems, we have also examined the Pd and W based multilayers that have been frequently studied in the literatures.

Firstly, Pd also has FCC crystal structure with crystal lattice about 3.88 Å. Therefore, based on our design strategy, the Pd/Co heterostructures on MgO (110) should also develop the tilted magnetic anisotropy and field-free SOT switching. To prove this hypothesis, we

have deposited two types of heterostructures, i.e. Pt(5)/Pd(1.4)/Co(0.6)/Pd(1.4)/NiO(20)/MgO(110) and Pt(5)/Pd(1.4)/Co(0.6)/Pd(1.4)/MgO(110) heterostructures, as shown in **Figure R14**. It is noted that the field-free SOT switching is observed in both heterostructures. Moreover, the dependences of the field-free SOT switching behaviors on crystalline orientations are consistent with the results in manuscript. These results highlight that our design strategy can be extended to other material systems.

Figure R14. The field dependence of anomalous Hall resistance and field-free SOT switching when current pulse is applied along $[\bar{1}10]$ and $[001]$ directions in (110) -oriented Pt(5 nm)/Pd(1.4 nm)/Co(0.6 nm)/Pd(1.4 nm)/MgO and Pt(5 nm)/Pd(1.4 nm)/Co(0.6 nm)/Pd(1.4 nm)/NiO(20 nm)/MgO heterostructures.

Moreover, as the referee suggested, we have also studied the W-based heterostructures with different stacking orders and thicknesses. **Figure R15** show representative results. Unfortunately, we find that the introducing of W/Co interface would greatly suppress the perpendicular magnetic anisotropy, as revealed by the field dependence of anomalous hall resistance in **Figure R15**. Therefore, the tilted magnetic anisotropy is not present. Note that the absence of perpendicular magnetic easy axis has been previously observed and is mainly ascribed to the magnetic dead layer on the W/Co interface (*J. Appl. Phys.* **126**, 133902 (2019); arXiv:2102.01283 (2021)).

Figure R15. The absence of remnant perpendicular magnetization in the W-based heterostructures.

In summary, the results in Pd-based multilayers demonstrate that our design strategy can be extended to other material systems. We have included these results on page 10 of revised manuscript and Fig. 25 of Supplementary Materials.

2. Moreover, I suggest that the authors should discuss the merits of their method compared to already reported ones showing field-free SOT switching using tilted magnetic anisotropy [Zhi Li, ACS Appl. Electron. Mater. 2022, H. Kim, Adv. Func. Mater. 2022, L. You, PNAS 2015].

Response:

We are particularly thankful for this important question. As the referee suggested, discussions on tilted anisotropy and comparison with other methods are now included on the page 10 of the revised manuscript and Table 1 of Supplementary Materials.

The referee has listed three main methods to induce titled anisotropy. We will compare our strategy with these methods in the following. Firstly, in the reference [L. You, PNAS 2015], the tilted anisotropy is induced by fabricating a wedge-like nanomagnet device with lateral size of a few hundred nanometers, which requires complex fabrication process. Secondly, in the reference [H. Kim, Adv. Func. Mater. 2022], the tilted anisotropy is induced by interlayer magnetic coupling in the multilayer structures, which demands a very precise control of thickness in each layer. In addition, the additional layers also impair the inherent advantages of a single layer magnet. Thirdly, in the reference [Zhi Li, ACS Appl. Electron. Mater. 2022], an easy cone anisotropy is induced, and the deterministic switching is induced

by the remnant in-plane magnetization component. However, due to the special type of anisotropy, the zero-field SOT switching ratio determined by ΔR_{xy} is very small (less than 2%) as shown in the reference [Zhi Li, ACS Appl. Electron. Mater. 2022]. Furthermore, none of these studies have carefully checked the endurance of SOT switching.

By contrast, in our proposed method by designing crystal symmetry and dislocation, we can realize the tilted anisotropy in the single-layer ferromagnetic film in the whole sample, without the use of complex fabrication or interlayer coupling. The field-free SOT switching is achieved with relatively low critical current density (10^{11} A/m²), high switching ratio (70%), and good endurance (10^4 cycles). These highlight the advantages of our method that could be beneficial for future applications. The important parameters are tabulated below:

Table 1. The comparison with other studies based on tilted magnetic anisotropy.

Heterostructure	Strategy	FM layer Number	Fabrication	Switching Ratio ($\Delta R_{xy}/R_{xy}$)	$J_c(\times 10^{11}$ A/m ²)	Endurance (cycle)	Ref
Ta/CoFeB/MgO	Device Asymmetry	1	Nano-fabrication (wedged nano-device of a few hundred nm)	~100%	2.5	2	[1]
Pt/[Gd/Co] ₉	Interlayer Coupling	9	As-deposited	N/A	3.2	15	[2]
CoPt	In-plane remnant magnetization	1	As-deposited	~2%	7.5	16	[3]
Pt/Co	Dislocation	1	As-deposited	~70%	3	10^4	This work

[1] You, L. et al. Switching of perpendicularly polarized nanomagnets with spin orbit torque without an external magnetic field by engineering a tilted anisotropy. *Proc. Natl. Acad. Sci. U. S. A.* **112**, 10310-10315 (2015).

[2] Kim, H. J. et al. Field-Free Switching of Magnetization by Tilting the Perpendicular Magnetic Anisotropy of Gd/Co Multilayers. *Adv. Funct. Mater.* **32**, 2112561 (2022).

[3] Li, Z. et al. Field-Free Magnetization Switching Induced by Bulk Spin–Orbit Torque in a (111)-Oriented CoPt Single Layer with In-Plane Remanent Magnetization. *ACS Appl. Electron. Mater.* **4**, 4033-4041 (2022).

3. Secondly, I wonder if the magnetization is fully switched by SOT without a magnetic field.

It seems odd to me that the change in Hall resistance (ΔR_{xy}) for the sample with $\theta_M = 16^\circ$ is much larger than that for the sample with $\theta_M \sim 90^\circ$. The authors should compare the field-free SOT switching results to the anomalous Hall resistance measured with perpendicular magnetic fields and show how much the magnetization is switched. I don't think it is reasonable to compare critical switching current density with other results unless full switching is achieved.

Response:

We thank the referee for pointing out this issue. Firstly, we want to clarify that the AHE resistance are different for the $\theta_M = 16^\circ$ and $\theta_M = 84^\circ$ heterostructures, due to the shunting current in additional Pt layers. As shown in the hysteresis loop of R_{xy} (red curves) in Fig. R16 and R17, the AHE resistance at zero field is approximately 0.25Ω and 0.17Ω for the $\theta_M = 16^\circ$ and $\theta_M = 84^\circ$ heterostructures.

By comparing the hysteresis loop of R_{xy} and current-driven SOT switching, we find that the field-free SOT switching ratio of heterostructure with $\theta_M = 16^\circ$ is about 70%, as shown in Fig. R16. For heterostructure with $\theta_M = 84^\circ$, measurements on multiple devices show that the switching ratio is about 40-60%. The incomplete switching could be ascribed to the pinning effect and the current shunting from the Hall voltage arms. Note that the switching ratio of R_{xy} is close to the reported values in other references [*Nat. Commun.* **10**, 233 (2019) (switching about 100%), *Phys. Rev. Appl.* **15**, 014017 (2021) (switching ratio about 30%), *Nat. Nanotechnol.* **16**, 277-282 (2021) (switching ratio about 20%), *Nat. Mater.* **20**, 800-804 (2021) (switching ratio about 70%), *Nat. Commun.* **12**, 6524 (2021) (switching ratio about 60%) and *Nat. Commun.* **13**, 3539(2022) (switching ratio about 50%).]. Therefore, the comparison of critical switching current densities with other references are reasonable.

Figure R16. The spin-orbit torque switching ratio of device with sample $\theta_M = 16^\circ$. The dash lines denote the remnant R_{xy} in the zero field.

Figure R17. The field-free SOT switching ratio of sample $\theta_M = 84^\circ$ in various devices. The current pulse is applied along the $[\bar{1}10]$ direction.

To further verify the SOT switching, we have examined the field-free SOT switching behaviors by using the polar-MOKE microscope. The results of these two samples are shown in Figure R18 below. The magnetization is almost fully switched.

Figure R18. The polar-MOKE images of field-free SOT switching. **a-b**, the results of sample $\theta_M = 16^\circ$ with current pulse of $3.6 \times 10^{11} \text{ A/m}^2$. **c-d**, the results of sample $\theta_M = 84^\circ$ with current pulse of $3.9 \times 10^{11} \text{ A/m}^2$. The pulse width is 2 ms. The initial magnetization direction was set by a magnetic field μ_0H , as shown in the upper Panel (for sample $\theta_M = 16^\circ$, μ_0H is 50 mT and along $[001]$. For sample $\theta_M = 84^\circ$, μ_0H is 10 mT and along $[110]$). The field-free SOT switching results are shown in lower panel. The current channel is $20 \mu\text{m}$ wide.

Below is a list of major changes:

- 1: The density and spatial extent of dislocations are characterized. This issue is mentioned on page 5 of revised manuscript. The results are included in Note 1 and Fig.5 of Supplementary Materials.
- 2: The issue of partial switching loop along [001] is clarified. The results are incorporated in Fig. 20 of Supplementary Materials.
- 3: The Joule heating effect in the SOT switching is characterized. This issue is mentioned on page 9 of revised manuscript. The results are included in Note 5, Fig. 19 of Supplementary Materials.
- 4: The field-free SOT switching ratio is characterized. This issue is mentioned on page 9 of revised manuscript. The results are included in Fig. 17 of Supplementary Materials.
- 5: The additional SOT switching experiments at different magnetic field are conducted. This issue is mentioned on page 10 of revised manuscript. The results are included in Note 6, Fig. 22 and Fig. 23 of Supplementary Materials.
- 6: The field-like torque and damping-like torque efficiencies are characterized. This issue is mentioned on page 9 of revised manuscript. The results are included in Note 4 and Fig. 18 of Supplementary Materials.
- 7: The absence of tilted magnetic easy axis and field-free SOT switching in (100)- and (111)-oriented heterostructures are confirmed. This issue is mentioned on page 9 of revised manuscript. The results are included in Fig. 21 of Supplementary Materials.
- 8: The pulse-width-dependent field-free SOT switching experiment is conducted. This issue is mentioned on page 9 of revised manuscript. The results are included in Fig. 20 of Supplementary Materials.
- 9: The accurate tilted angle θ_M of samples for SOT switching is clarified in the revised manuscript.
- 10: The applicability of the present method is examined in other materials systems. This issue is mentioned on page 10 of revised manuscript. The results are included in Fig. 25 of Supplementary Materials.
- 11: The merits of this work are compared with literatures based on tilted magnetic anisotropy, mentioned on page 10 of revised manuscript. The discussion is included in Note 7 and Table 1 of Supplementary Materials.

Reviewers' Comments:

Reviewer #1:

Remarks to the Author:

Liang, et al. has substantially revised the manuscript according to the comments and questions from referees. They have added (1) the (100) and (111) control samples to show the uniqueness of the (110) substrates, (2) the SOT efficiency characterizations, (3) analysis of the heating effect and (4) larger range estimation of the dislocation density, which can ease the reproducing complexity of others and guide the later use of this technique. However, I am still questioning the buildup of the tilted PMA easy axis by the M-H loop or MOKE measurement. Logically, if the easy axis of a PMA film is tilted from the film normal (the nominal one), its magnetization behavior should be in principle measurable following the coming logic. A tilted magnetization can be deemed equivalently in some sense as the chiral coupling between an in-plane one and a PMA one. In this case, an in-plane bias field (a fixed small magnetic field applied in-plane) should influence the magnetizing process of the PMA component by Hz. And moreover, this influence should be maximized (totally absent) as the applied in-plane field is colinear (perpendicular) with the direction of the in-plane component. In this case, one would expect a similar magnetization behavior as observed in an interlayer-DMI-coupled system [Han D. S. et al. Long-range chiral exchange interaction in synthetic antiferromagnets. *Nat. Mater.* 2019, 18, 703–708.].

Here should be Figure 2 of the reference paper

Fig.1 Abnormal magnetization behavior of an interlayer-DMI coupled system whose perpendicular magnetization process M_z -Hz (a) can be affected by an in-plane field H_{in} as H_{in} is applied in a proper direction (b and c).

This measurement needs both Hz and Hx, which is not so unique. I think the manuscript should take this measurement into account to straightforwardly evidence the existence of the tilted PMA easy axis for the (110) system but absent for the (100) and (111) counterparts.

Reviewer #2:

Remarks to the Author:

I appreciate the authors' efforts to responding to the reviewers' comments with additional experiments and explanations. This alleviates my concerns and the revised manuscript is much improved. Therefore, I recommend this manuscript for publication in Nature Communications.

Reviewer #1:

Liang, et al. has substantially revised the manuscript according to the comments and questions from referees. They have added (1) the (100) and (111) control samples to show the uniqueness of the (110) substrates, (2) the SOT efficiency characterizations, (3) analysis of the heating effect and (4) larger range estimation of the dislocation density, which can ease the reproducing complexity of others and guide the later use of this technique.

Response: We thank this referee's for recognizing our revised manuscript being "ease the reproducing complexity of others and guide the later use of this technique". Following her/his insightful suggestions, we have now performed additional experiments and analyses in this revised manuscript and supplementary materials. Below we present our responses to specific comments.

1. However, I am still questioning the buildup of the tilted PMA easy axis by the M-H loop or MOKE measurement. Logically, if the easy axis of a PMA film is tilted from the film normal (the nominal one), its magnetization behavior should be in principle measurable following the coming logic. A tilted magnetization can be deemed equivalently in some sense as the chiral coupling between an in-plane one and a PMA one. In this case, an in-plane bias field (a fixed small magnetic field applied in-plane) should influence the magnetizing process of the PMA component by Hz. And moreover, this influence should be maximized (totally absent) as the applied in-plane field is colinear (perpendicular) with the direction of the in-plane component. In this case, one would expect a similar magnetization behavior as observed in an interlayer-DMI-coupled system [Han D. S. et al. Long-range chiral exchange interaction in synthetic antiferromagnets. Nat. Mater. 2019, 18, 703–708.].

Here should be Figure 2 of the reference paper.

Fig.1 Abnormal magnetization behavior of an interlayer-DMI coupled system whose perpendicular magnetization process M_z -Hz (a) can be affected by an in-plane field H_{in} as H_{in} is applied in a proper direction (b and c).

This measurement needs both Hz and H_x , which is not so unique. I think the manuscript should take this measurement into account to straightforwardly evidence the existence of the tilted PMA easy axis for the (110) system but absent for the (100) and (111) counterparts.

Response:

We thank the referee for making this insightful comment. As the referee suggested, we

have confirmed that the in-plane bias field H_{IP} indeed affects the perpendicular magnetization switching (M_z - H_z) in our heterostructures, which thereby further confirmed the tilted magnetic anisotropy. Here, the M_z component is detected by the polar MOKE as sweeping out-of-plane field H_z . As shown in **Figure R1a** and **1c**, when the H_{IP} is applied along $[001]$ direction, the hysteresis loops of polar MOKE signal as a function of H_z show a clear shift, for (110) -oriented heterostructures with $\theta_M=84^\circ$ and $\theta_M=16^\circ$. Moreover, the loop shift changes to the opposite direction as we reverse the H_{IP} direction. By contrast, the loop shift is absent when the H_{IP} is applied along $[\bar{1}10]$ direction, as shown in **Figure R1b** and **1d**. These results further verify the existence of the tilted magnetic easy axis in our (110) -oriented heterostructures.

Figure R1. The shift of polar MOKE hysteresis loop induced by in-plane bias field H_{IP} in (110) -orientated heterostructures. **a** and **b** show the polar MOKE hysteresis as sweeping out-of-plane field H_z of (110) -oriented heterostructure with $\theta_M=84^\circ$, when H_{IP} is applied along **a**, $[001]$ or **b**, $[\bar{1}10]$ direction. **c** and **d** show the polar MOKE hysteresis as sweeping out-of-plane field H_z of (110) -oriented heterostructure with $\theta_M=16^\circ$, when H_{IP} is applied along **c**, $[001]$ or **d**, $[\bar{1}10]$ direction. The upper panels show the schematic diagrams of H_{IP} and tilted magnetic easy axis. The magnitude of H_{IP} is 10 mT.

As shown in **Figure R2**, the shift of polar MOKE hysteresis loop is always absent in the (100) -oriented and (111) -oriented heterostructures, when H_{IP} is applied. These results are consistent with the absence of tilted magnetic easy axis in (100) -oriented and (111) -oriented heterostructures.

Figure R2. The absence of shift of polar MOKE hysteresis loop with in-plane bias field H_{IP} in (100)- and (111)-oriented heterostructures. **a** and **b** show the polar MOKE hysteresis as sweeping out-of-plane field H_z of (100)-oriented heterostructure, when H_{IP} is applied along **a**, [001] or **b**, [010] direction. **c** and **d** show the polar MOKE hysteresis as sweeping out-of-plane field H_z of (111)-oriented heterostructure, when H_{IP} is applied along **c**, $[11\bar{2}]$ or **d**, $[\bar{1}10]$ direction. The upper panels show the schematic diagrams of H_{IP} and crystalline directions. The magnitude of H_{IP} is 10 mT.

This issue is mentioned on page 7 of revised manuscript. The results are included in Fig.12 and Fig.13 of Supplementary Materials.

Reviewer #2:

I appreciate the authors' efforts to responding to the reviewers' comments with additional experiments and explanations. This alleviates my concerns and the revised manuscript is much improved. Therefore, I recommend this manuscript for publication in Nature Communications.

Response:

We thank this referee for now in the position of supporting the publication of our manuscript in Nature Communications.

Reviewers' Comments:

Reviewer #1:

Remarks to the Author:

The authors have properly replied to my last concern and by a specific magnetization measurement (similar with the interlayer DMI measurement) they confirm the tilted PMA anisotropy for the proper substrate orientations. In current stage, I would like to recommend its publication.

Aug. 15th, 2023

Subject: Manuscript NCOMMS-22-41247C, by Y. Liang, *et al.*

Dear Editor,

Thank you very much for your kind letter dated on **Jul. 27, 2023** concerning our manuscript titled “Field-free Spin-orbit Switching of Perpendicular Magnetization Enabled by Dislocation-induced In-plane Symmetry Breaking” (NCOMMS-22-41247B). We really appreciate the referees’ valuable and helpful comments. Following their stimulating comments, we have revised our manuscript accordingly and made necessary changes which were marked in **blue**. We believe that we have clarified all concerns from referees, both the clarity and robustness of our manuscript have been enhanced. This revised manuscript could attract broad interest from the spintronic community, and meet the publishing criteria of Nature Communications.

Below is a list of major changes:

1: The manuscript has been checked and revised according to the guidance from author checklist.

Yours Sincerely

Yuan-Huan Lin on behalf of all coauthors.

Reviewer #1:

The authors have properly replied to my last concern and by a specific magnetization measurement (similar with the interlayer DMI measurement) they confirm the tilted PMA anisotropy for the proper substrate orientations. In current stage, I would like to recommend its publication.

Response: We thank this referee for now in the position of supporting the publication of our manuscript in Nature Communications.

In summary, we believe that all comments from referees have been satisfactorily addressed. We hope that these revisions will now make this manuscript suitable for publication in *Nature Communications*.

Best regards!

Yuan-Hua Lin on behalf of all coauthors